# Validation of quantitative assessment of indocyanine green fluorescent imaging in a one-vessel model

**Anna Duprée**[1]*, **Henrik C. Rieß**[2], **Philipp H. von Kroge**[1], **Jakob R. Izbicki**[1], **Eike S. Debus**[2], **Oliver Mann**[1], **Hans O. Pinnschmidt**[3], **Detlef Russ**[4], **Christian Detter**[5], **Sabine H. Wipper**[2]

1 Department General, Visceral and Thoracic Surgery, University Medical Center Hamburg-Eppendorf, Hamburg, Germany, 2 Department of Vascular Medicine, University Heart Center, University Medical Center Hamburg-Eppendorf, Hamburg, Germany, 3 Department of Medical Biometry and Epidemiology, University Hospital Eppendorf, Hamburg, Germany, 4 Department for the Development of Applications, Institute for Laser Technology, University Ulm, Ulm, Germany, 5 Centre of Cardiology and Cardiovascular Surgery, University Hospital Eppendorf, Hamburg, Germany

* adupree@uke.de

**Data Availability Statement:** All relevant data are provided in the Figshare repository at: 10.6084/m9.figshare.13011263.

## Abstract

### Objectives

Evaluation of intestinal perfusion remains subjective and depends on the surgeon´s individual experience. Intraoperative quality assessment of tissue perfusion with indocyanine green (ICG) fluorescence using a near-infrared camera system has been described in different ways and for different indications. The aim of the present study was to evaluate fluorescent imaging (FI) in the quantitative assessment of intestinal perfusion in a gastric tube model in pigs and to compare the results to results obtained with florescent microspheres (FM), the gold standard for tissue perfusion.

### Methods

Seven pigs (56.0±3.0 kg), both males and females, underwent gastric tube formation after transection and ligation of the gastric arteries, except the right gastroepiploic artery, to avoid collateral blood flow.

After baseline assessment (T0), hypotension (T1) was induced by propofol (Karampinis et al 2017) (< 60 mmHg). Then, propofol was paused to obtain normotension (T2, Mean arterial pressure (MAP) 60–90 mmHg). Finally, hypertension (T3, MAP>90 mmHg) was induced by norepinephrine.

Measurements were performed in three regions of interest (ROIs) under standardized conditions: the fundus (D1), corpus (D2), and prepyloric area (D3). Hemodynamic parameters and transit-time flow measurement (TTFM) in the right gastroepiploic artery were continuously assessed. FI, FM and the partial pressure of tissue oxygen (TpO$_2$) were quantified in each ROI.

**Funding:** The authors received no specific funding for this work.

**Competing interests:** This research did not receive any specific grant from funding agencies in the public, commercial, or not-for-profit sectors. Cost of animals and equipment were covered by a department internal fund, staple devices were provided by Medtronic GmbH Germany. This does not alter our adherence to PLOS ONE policies on sharing data and materials.

## Results

The study protocol could successfully be performed during stable hemodynamics. Flow in the gastroepiploic artery measured by transit time flow measurement (TTFM) was related to hemodynamic changes between the measurements, indicating improved blood flow with increasing MAP. The distal part of the gastric tube (D1) showed significantly ($p<0.05$) impaired perfusion compared to the proximal parts D3 and D2 using FM. ICG-FI also showed the highest values in D3 and the lowest values in D1 at all hemodynamic levels (T1-T3; $p<0,05$).

## Conclusion

Visual and quantitative assessment of gastric tube perfusion is feasible in an experimental setting using ICG-FI. This might be a promising tool for intraoperative assessment during visceral surgery in the future.

## Introduction

The assessment of organ perfusion is a crucial step in several types of visceral surgery, such as esophagectomy, colonic resection, and surgery for mesenteric ischemia. The quality of tissue perfusion is generally predicted by serosal color, grade of peristalsis, pulsation, and bleeding from tissue margins. Nevertheless, clinical evaluation is highly subjective and varies with the individual capability of the surgeon [1]. Independent and quantitative methods to gage visceral perfusion are of mixed effectiveness. Indocyanine green (ICG) is an appropriate contrast agent for fluorescent imaging (FI) [2]. It is a safe fluorescent dye that has been used for decades in the measurement of liver function, via ICG clearance, and microcirculation and in ophthalmic angiography. It is quickly and completely eliminated by the liver with a half-life of 2.4 minutes, which allows multiple applications. There have only been a few adverse events reported, and side effects other than an iodine allergy are unknown [3–5]. ICG shows an absorption maximum at 805 nm and an emission maximum at 830 nm [3]. Because of its user-friendly properties, FI has enjoyed growing popularity in the assessment of tissue perfusion during surgery due to its suspected influence on anastomotic healing [6, 7]. Nevertheless, the technique and its interpretation remain subjective due to the mere visual analysis of the fluorescence during the operation. Therefore, the advantage of using ICG-FI in daily routine practice is currently uncertain [7].

Detter et al. described a quantitative assessment of myocardial perfusion during graded coronary stenosis by analyzing ICG fluorescence [8] using an ICG-based imaging system [9]. To date, only poor quantitative assessment strategies have been used in visceral surgery.

The aim of this study was to evaluate and validate ICG-FI in visceral surgery for visual and quantitative assessment of tissue perfusion. For this purpose, we developed an experimental in vivo single-vessel model using gastric tube formation, as it is done during esophagectomy in pigs, for standardized assessment of tissue perfusion. Measurement of tissue perfusion by ICG-FI was compared to fluorescent microspheres (FM), the experimental gold standard.

## Methods

This study was carried out in strict accordance with the recommendations in the Guide for the Care and Use of Laboratory Animals of the National Institutes of Health. The protocol was

approved by the Committee on the Ethics of Animal Experiments of the Authority for Health and Consumer Protection Hamburg (Protocol Number: 113/14, S1 File). All surgery was performed under deep anesthesia, and all efforts were made to minimize suffering.

The experiments were performed in 7 pigs, both males and females, weighing 56.0 +/- 3.0 kg, at the Institute for Surgical Research, University Center Hamburg-Eppendorf (Hamburg, Germany). All animals were provided by the pig farm KG Fokken in Schmalfeld, Germany. To get used to the new environment, the animals were delivered at least 7 days before the experiments. The animals were never kept alone to reduce stress. All shelters were fitted with toys, and the animals had open air access at least once a day. Care and feeding were provided by the team of the Institute for Surgical Research. Since these were final experiments, postoperative care was not necessary. The animals were sacrificed under deep anesthesia.

After intramuscular premedication with azaperone (4 mg/kg), midazolam (0.3 mg/kg), ketamine (5 mg/kg), and atropine sulfate (0.15 mg/kg), intravenous anesthesia was induced by propofol (0.06 mg/kg) and maintained by continuous infusion of fentanyl (0.01 mg/kg/h), midazolam (0.1 mg/kg/h), ketamine (0.1 mg/kg/h), and propofol (3 mg/kg/h). The animals were endotracheally intubated and received pressure-controlled ventilation at 15 cm $H_2O$, with a positive end-expiratory pressure of 7 cm $H_2O$ at 16 breaths per minute using 30% oxygen. Heparin (400 U/kg) was administered to achieve an activated clotting time of at least 300 seconds. Administrations were repeated every 3 hours.

A 6F arterial catheter was introduced into the right carotid artery for continuous blood pressure monitoring. A central venous catheter was inserted into the right jugular vein for delivery of infusions and heparin. A 4F arterial catheter with an embedded thermistor was inserted into the right femoral artery for continuous hemodynamic monitoring of stroke volume, cardiac output, cardiac index, systemic vascular resistance, and arterial blood pressure by determination of thermodilution. These parameters were documented with a PiCCO device (Pulsion Medical Systems, Munich, Germany) on the basis of arterial pulse contour analysis. For application of the FM, a left atrial catheter was placed, and a pigtail catheter was inserted into the abdominal aorta via the right femoral artery for withdrawal of reference blood samples with a known, constant withdrawal rate to calculate the rate of blood flow per tissue sample.

## Surgical procedures

Midline laparotomy was performed. A catheter was placed into the bladder for urinary drainage. Gastric tube formation was performed under ligation of all arteries, except the gastroepiploic artery and its arcade. Dissection of the minor curvature was performed using Endo-GIA (Covidien, black cartridge), resulting in a gastric tube that was 3 cm in diameter. In this way, the perfusion was dependent on one vessel, leading to a standardized "one-vessel model". Three regions of interest (ROIs) were defined as follows: the fundus (D1), corpus (D2), and prepyloric area (D3).

Thereafter, flexible polarographic measuring probes (Licox, Germany) for continuous measurement of the tissue oxygen tension (partial pressure of tissue oxygen [tpO$_2$]) were inserted in the proximal (D3) and distal end of the tube (D1). For the continuous determination of blood flow, a Doppler flow probe (CardioMed Flowmeter, Medi-Stim AS, Oslo, Norway) was placed around the gastroepiploic artery. The abdomen was covered particularly to reduce loss of fluid and temperature.

## Fluorescent microspheres

FM that were 15 μm in diameter (Molecular Probes, Eugene, Ore) were used for quantitative assessment of gastric tube perfusion in ml/min/g during each measurement. Microspheres

labeled with 7 different fluorescent colors were randomly selected for application. The excitation and emission wavelengths for each of the fluorescent microspheres were as follows: blue, 356/424 nm; blue-green, 427/468 nm; yellow-green, 495/505 nm; orange, 534/554 nm; red, 570/598 nm; crimson, 612/638 nm; and scarlet, 651/680 nm. They did not interfere with the excitation and emission wavelengths of ICG. FM were placed in an ultrasonic water bath for 5 to 10 minutes to disperse the microspheres; then, they were vortexed twice for 3 minutes to ensure proper mixing before injection. Approximately $3,33 \times 10^6$ FM were suspended in physiological saline solution to a volume of 10 mL and injected continuously over 60 seconds into the left atrium at each measurement. Reference blood samples were withdrawn in anticoagulated (5 mL of 3.13% sodium citrate) syringes with a constant-rate withdrawal pump at 3.18 mL/min over 3 minutes. FM injection was started when the withdrawn blood reached the suction syringe. At the end of the experiments, the gastric tube was excised and fixed in 10% formaldehyde solution for at least 8 days. Thereafter, it was dissected according to a standardized protocol into 28 tissue pieces with a mean weight of 3.5±0.2 g. The specimens were processed for determination of blood flow by spectrofluorometry according to the standard method described by Glenny et al. [10].

## Fluorescence intensity

ICG-FI was performed with the FCI system (LLS GmbH) developed by the Institute of Laser Technology of the University of Ulm [11]. ICG was administered through a peripheral venous line. The dose of ICG was adapted to body weight (0.02 mg/kg body weight). The area of interest was illuminated with near-infrared light with a wavelength of 785 nm that was provided by infrared laser diodes with a total output of 80 mW in a field of view that was 10 cm in diameter. The fluorescent emission of the excited dye was detected by an infrared-sensitive charge-coupled device camera system equipped with a bandpass filter for the selective transmission of light at a central wavelength of 830 nm. The dynamic range of the camera was 54 dB. The signals of the camera were digitized with a frame grabber card that provided a resolution of 8 bits. Images were acquired at a rate of 25 frames per second and recorded in real time. The camera distance was measured and calibrated after each measurement at a distance of ≈25 cm. The FCI images were displayed in real time on a computer monitor and analyzed using a digital image processing system with a temporal resolution of 20 ms and spatial resolution of ≈0.2 mm at a penetration depth of 4 mm (LLS GmbH).

## Image and data analysis

ICG-FI data analysis was performed as previously described in detail [8]. Briefly, the corresponding time-dependent fluorescence intensity curves in each region of interest (ROI) were analyzed. The mean value and standard deviation (SD) were calculated for each ROI in a sequence of 60 seconds after the injection of ICG. To distinguish gastric perfusion, 3 different parameters derived from the time-dependent fluorescent curve were defined (Fig 1):

**Background-Subtracted Peak Fluorescence Intensity (BSFI).** To calculate the BSFI from the time-dependent fluorescence intensity curves, the initial intensity value before the injection of ICG was subtracted from the peak fluorescence intensity during the first passage of the dye through the gastric tube [8].

**Slope of Fluorescence Intensity (SFI).** This parameter is represented by the maximal slope during the increase of the time-dependent fluorescence intensity induced by the first wave of the dye that passes through the capillaries of the gastric tissue.

**Time to slope (TTS).** TTS is defined as the time between injection and first detection, represented by the first increase in the fluorescence intensity signal in the ROI.

## ICG-FI data analysis

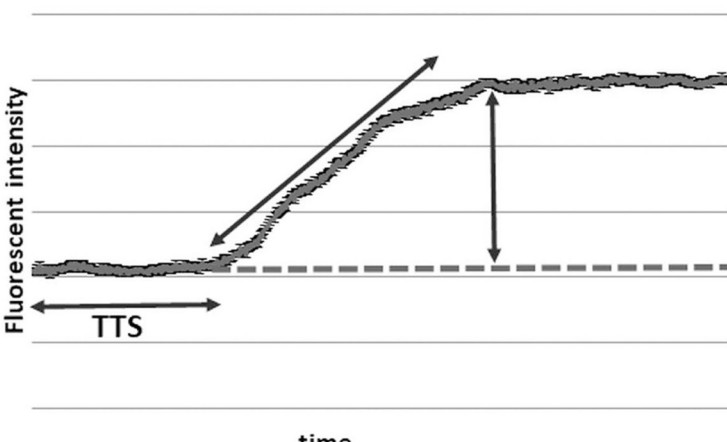

**Fig 1. ICG-FI data analysis.** ICG-FI data are analyzed in the corresponding fluorescence intensity curve of each ROI. ICG-FI: Indocyanine green fluorescent imaging; ROI: region of interest; BSFI: Background subtracted fluorescent intensity; SFI: Slope of fluorescent intensity; TTS: Time to slope.

Each ICG-FI sequence was recorded online for 120 seconds with real-time digitizing. BSFI, SFI and TTS data were analyzed offline with a customized software package (LLS GmbH).

### Experimental protocol

After instrumentation, the hemodynamic, metabolic, and tissue oxygenation parameters were determined according to a standardized protocol. ICG-FI and TTFM were performed, and FM were quantified during each measurement. Heart rate, systemic arterial pressure, cardiac output, central venous pressure, pulmonary arterial pressure, systemic vascular resistance and the signals generated by the TTFM probe around the gastric epiploic artery were recorded continuously. Tissue oxygenation was continuously determined by inserting flexible measuring probes (Licox, Germany) into the ROI and captured for each measurement.

After the baseline assessment (T0), blood pressure was modified by norepinephrine infusion to obtain 3 different mean arterial pressure [12] levels: hypotension (MAP < 60 mmHg, T1), normotension (MAP 60–90 mmHg, T2), and hypertension (MAP > 90 mmHg, T3). Hypotension was reached by application of a propofol bolus, while hypertension was induced using norepinephrine. Every time the target MAP was reached, a 30-minute stabilization period was observed.

After finishing the measurements, the animals were sacrificed by a veterinarian using T61 under deep anesthesia.

### Histopathology

Representative specimens of the ROIs were assessed and stored in 3.5% buffered formalin. All samples were routinely processed and embedded in paraffin, and 5-μm slices were stained with hematoxylin and eosin. The slices were examined by an experienced pathologist (blinded to the treatment) using an established scoring system [13]. Mucosa taken from the anterior gastric corpus was divided into the following grades under a light microscope: grade 0: normal gastric mucosa; grade 1: surface mucosa cells were damaged; grade 2: in addition to extensive

luminal damage, cells lining the gastric pits were also disrupted and exfoliated; grade 3: cell destruction extended into the gastric gland.

## Statistics

Continuous variables are presented as the means and standard deviations (SD) across animals. Data distributions were assessed using histograms and box plots. The variables BSFI, SFI and TTS were inverse transformed prior to further analyses to minimize skewness and heteroscedasticity. Data were then subjected to mixed model analyses to take repeated measurements within animals into account. For the dependent variables BSFI, SFI and TTS, we fitted initial models containing fixed effects of area, time, FM and area-by-time. Random intercepts for the raters and animals were fitted, and repeated measures over time with area-by-animal-by-rater interactions were accounted for. For the dependent variables $tpO_2$ and FM, fixed effects of area, time and area-by-time were tested, accounting for repeated measures over time with area-by-animal interactions. For the dependent variable use of FM, random intercepts for animals were fitted, but they were not for the variable $tpO_2$, as the final Hessian matrix was not definitely positive for the corresponding model. For the dependent variable "flow" and the hemodynamic dependent variables CO, CI, HR, MAP, SVR, GEDV and GVD, a fixed effect for time was fitted, and repeated measures over time within animals were accounted for. Non-significant fixed effect terms were excluded from the models using a hierarchical stepwise-backwards approach. For repeated measures, an autoregressive covariance structure was employed, while a variance components covariance structure was employed for estimating random intercepts. Time was treated as a categorical variable. Model-estimated marginal means with 95% confidence intervals are presented for individual dependent variables. The marginal means and 95% CIs were back-transformed into the original scales for graphic representation if applicable. A value of $p < 0.05$ was considered statistically significant. Statistical analysis was performed with the SPSS 23.0 statistical software package (SPSS Inc, Chicago, Ill).

## Results

High-quality ICG-FI images of gastric tube perfusion were obtained in all animals. All hemodynamic parameters were successfully documented and are summarized in Table 1. Flow in the gastroepiploic artery, measured by TTFM, is related to hemodynamic changes, indicating improved blood flow with increasing MAP.

### Quantitative assessment

The impairment of perfusion was quantified by FM and ICG-FI.

**Assessment of tissue perfusion with FM.** Gastric tube perfusion in ml/min/g assessed by FM is shown in Fig 2. The distal part of the gastric tube (D1) showed significantly ($p < 0.05$) impaired perfusion (T1: 1.04±0.45 ml/min/g; T2: 0.92±0.23 ml/min/g; T3: 1.12±0.45 ml/min/g) compared to the proximal parts D3 (T1: 1.12±0.41 ml/min/g; T2: 1.20±0.35 ml/min/g; T3: 1.45±0.37 ml/min/g) and D2 (T1: 1.12±0.46 ml/min/g; T2: 1.14±0.31 ml/min/g; T3: 1.51±0.59 ml/min/g) during all measurements. Hypertension (T3) led to slightly but not significantly improved perfusion in all ROIs of the gastric tube compared to hypotension and normotension (T1 and T2, respectively).

**Assessment of tissue perfusion by ICG-FI.** Gastric tube perfusion was successfully quantified using SFI, BSFI, and TTS (Fig 3).

*SFI*. The SFI (Fig 3a) showed significant impairment of microperfusion in the distal area (D1) of the gastric tube during all measurements (T1: 1.72E+12±8.43+11; T2: 2.08E+12±1.36

**Table 1. Hemodynamic parameters.**

| | T0 Baseline | T1 MAP <60 mmHg | T2 MAP 60–90 mmHg | T3 MAP >90 mmHg | p |
|---|---|---|---|---|---|
| CO (l/min) | 5.3 ± 1.3 | 5.5 ± 1.5 | 5.3 ± 1.3 | 4.9 ± 1.2 | n.s. |
| CI (l*min$^{-1}$*m$^{-2}$) | 3.6 ± 1.0 | 3.6 ± 1.1 | 3.8 ± 1.1 | 3.4 ± 0.9 | n.s. |
| HR (bpm) | 106.5 ± 34.1 | 111.5 ± 33.3 | 133.4 ± 24.1 | 134.0 ± 23.8 | <0.05 |
| MAP (mmHg) | 59.7 ± 14.3 | 51.4 ± 4.3 | 79.6 ± 5.5 | 101.5 ± 9.4 | <0.05 |
| SVR (dyn*s*cm$^{-5}$) | 930.0 ± 378.0 | 713.8 ± 154.9 | 1152.5 ± 237.6 | 1647.5 ± 451.8 | <0.05 |
| GEDV [14] | 650.8 ± 87.7 | 646.4 ± 69.0 | 661.4 ± 114.1 | 649.0 ± 124.7 | n.s. |
| CVP (cmH$_2$O) | 7.3 ± 1.5 | 6.8 ± 1.5 | 6.1 ± 1.4 | 5.6 ± 1.3 | <0.05 |

CO, cardiac output; CI, cardiac index; HR, heart rate; MAP, mean arterial pressure; SVR, systemic vascular resistance; GEDV, global end-diastolic volume; CVP, central venous pressure; hemodynamics were stable throughout the experiment, and heart rate, mean arterial pressure, systemic vascular resistance, and central venous pressure differed significantly between measurements (p<0.05).

+12; T3: 2.08E+11±1.64+12). The pyloric region (D3) showed enhanced FI (T1: 5.14E+12 ±3.09+12; T2: 7.43E+12±2.95E+12; T3: 8.64E+12±5.12+12) with significantly increased SFI levels compared to the distal area (D1) during all measurements (p<0.001). The SFI was also significantly different between D2 (T1: 3.04E+12±2.40E+12; T2: 4.53E+12±2.14E+12; T3: 4.41E+12±2.19E+12) and D3 (p<0.001). Hypertension (T3) and normotension (T2) resulted in no significant changes in microperfusion using SFI, while hypotension led to significantly worse perfusion in all areas (p<0.05).

*BSFI.* BSFI (Fig 3b) showed similar results to the SFI. Microperfusion in D1 was reduced using BSFI during all measurements (T1: 9.97E+14±5.69E+14; T2: 1.06E+15±5.90E+14; T3:

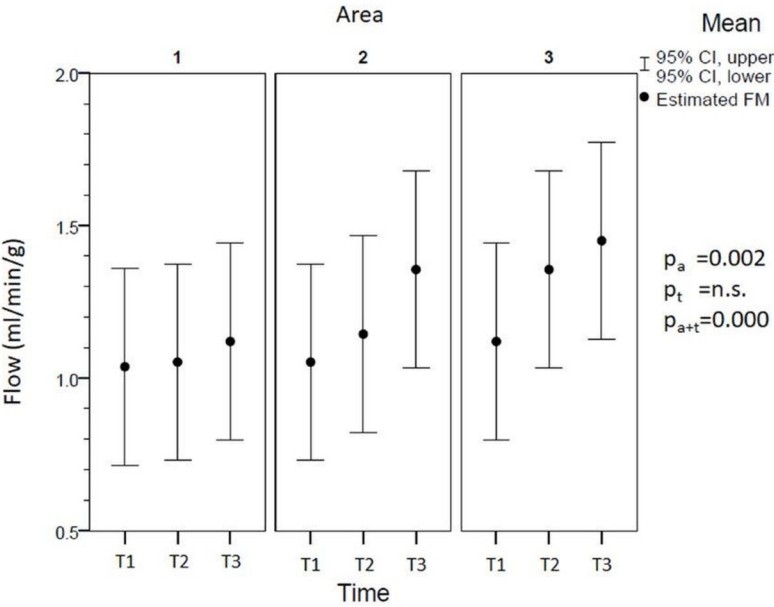

**Fig 2. Assessment of gastric perfusion using FM.** Flow in the 3 different areas of the gastric tube (D1-D3) using fluorescent microspheres (FM) as the gold standard for experimental assessment of microperfusion in ml/min/g tissue. P-values for the fixed effects reached significance for area (p$_a$) and time-dependent area (p$_{a*t}$), while time (p$_t$) alone remained insignificant (n.s.). Pairwise comparisons based on estimated marginal means with a confidence interval of 95% were significant between areas D1 and D2 at all time points (p<0.05).

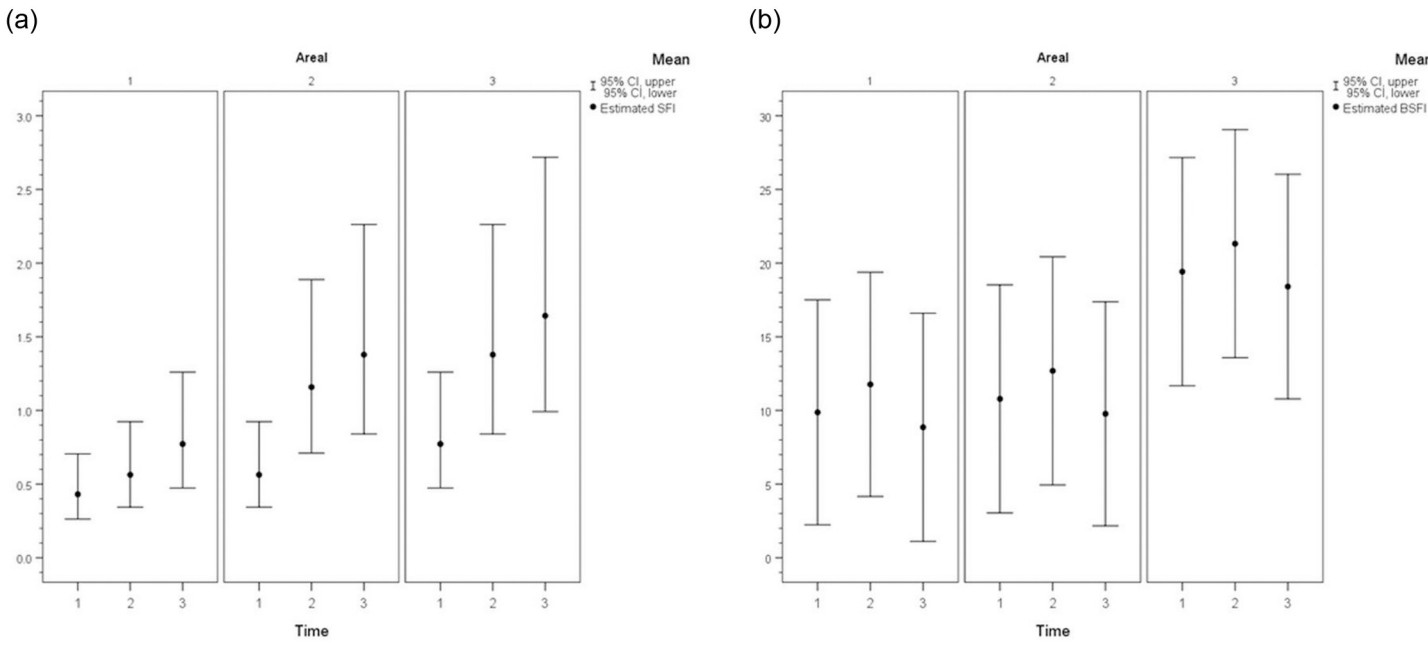

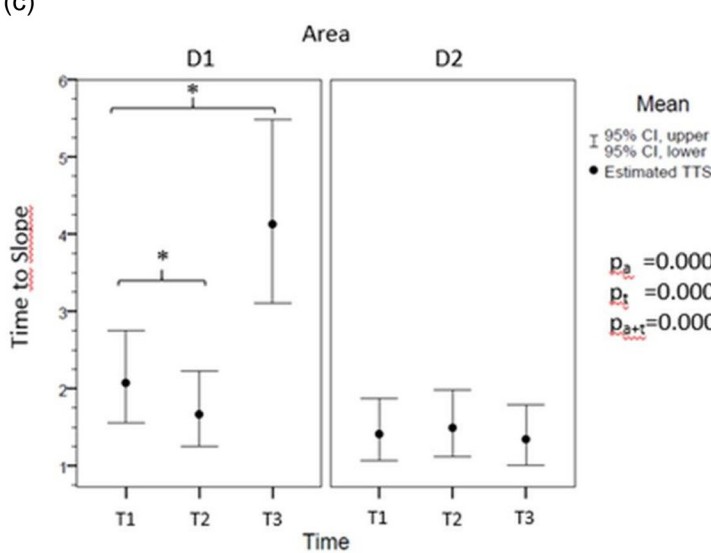

**Fig 3.** A. Assessment of gastric perfusion using ICG-FI (SFI). Assessment of gastric tube perfusion using ICG-fluorescent imaging (ICG-FI) using slope of fluorescence intensity (SFI) analysis. This parameter is represented by the maximal slope during the increase in time-dependent fluorescence intensity induced by the first wave of the dye that passes through the capillaries of the gastric tissue. The SFI showed reduced perfusion in D1. The perfusion difference between T1 (hypotension) and T2 (normotension) reached significance, as it did between areas D2 and D3 (*: p<0.05). P-values for fixed effects were significant for area ($p_a$) and time ($p_t$) but not for time-dependent area ($p_{a*t}$). The correlation to the FM results was also significant (p = 0.021). Pairwise comparisons were based on estimated marginal means with a confidence interval of 95%. The graph shows backtransformed data. B. Assessment of gastric perfusion using ICG-FI (BSFI). Assessment of gastric tube perfusion using ICG-fluorescent imaging (ICG-FI) using baseline subtracted fluorescence intensity (BSFI) analysis. To calculate the BSFI from the time-dependent fluorescence intensity curves, the initial intensity value before the injection of ICG was subtracted from the peak fluorescence intensity during the first passage of the dye through the gastric tube. BSFI showed reduced perfusion in D1. There were no significant perfusion differences between the time points T1 and T3 in any of the areas. P-values for fixed effects were significant for area ($p_a$) but not for time-dependent area ($p_{a*t}$) or time alone (n.s.). The correlation with the FM results was not significant (p = 0.064). Pairwise comparisons were based on estimated marginal means with a confidence interval of 95%. The graph shows backtransformed data. C. Assessment of gastric perfusion using ICG-FI (TTS). Assessment of gastric tube perfusion using ICG fluorescent imaging (ICG-FI) and time-to-slope (TTS) analysis. TTS is defined as the time between injection and first detection, represented by the first increase in the fluorescence intensity signal in the ROI. Data are shown as the ratio to D3 (radix of artery). D1 shows a significant delay in the fluorescence signal at all time points (*: p<0.05). P-values for fixed effects were significant for area ($p_a$), time ($p_t$), and time-dependent area ($p_{a*t}$). D2 showed no significance (n.s.). Pairwise comparisons were based on estimated marginal means with a confidence interval of 95%. The graph shows backtransformed data.

9.85E+14±5.11E+14). In contrast to the SFI results, the BSFI in D2 (T1: 1.20E+15±6.78E+14; T2: 1.14E+15±4.40E+14; T3: 1.15E+15±5.55E+14) had perfusion results similar to those in D1 (n.s.). D3 showed significantly better perfusion using BSFI during all measurements (T1: 1.96E+15±6.75E+14; T2: 2.28E+15±7.82E+14; T3: 2.03E+15±8.09E+14) than D2 and D1 ($p < 0.001$). BSFI was not significantly influenced by the hemodynamic changes during the measurements.

SFI showed a significant correlation with FM ($p = 0.021$), the gold standard, while BSFI was not significantly correlated with FM ($p = 0.067$).

*TTS*. The TTS ratio (Fig 3c) showed the later onset of florescence in FI in the distal parts (D1, D2) than in the pyloric region (D3). Thus, perfusion onset in D1 (T1: 2.47±1.81; T2: 1.77±0.58; T3: 4.48±1.94) was significantly ($p < 0.05$) delayed throughout the measurements. Hypertension (T3: 4.48 ±1.94), as well as hypotension (T1: 2.47±1.81), led to a significant additional retardation in D3 compared to D2 (T1: 1.47±0.51; T3: 1.43±0.49; $p < 0.001$).

**Histology.** As shown in Fig 4, the grade of mucosal injury increased significantly ($p < 0.05$) towards the distal part of the gastric tube (D1), indicating loss of epithelium integrity caused by impaired perfusion.

**Assessment of tissue oxygenation.** The tissue oxygenation shown in Fig 5 was significantly impaired in D1 (T1: 19.98±11.85; T2: 24.83±14.49; T3: 16.40±9.91) compared to D2 (T1: 30.08±5.20; T2: 37.11±13.73; T3: 38.14±15.28) and D3 (T1: 27.95±14.64; T2: 30.47±17.27; T3: 26.60±13.30) ($p < 0.05$).

## Discussion

There are various methods that address the assessment of intraoperative tissue perfusion. Of these, ICG fluorescence appears to be the most encouraging [1, 15]. Various studies report the usage of the technique in esophageal [16–19] and colorectal surgery [7, 20–27]. However, most publications report a visual assessment of fluorescence intensity. For example, a study in 40 esophagectomies showed that using ICG-FI is helpful in visualizing gastric tube perfusion prior to reconstruction. Thus, the method is found to be beneficial for the decision of whether the anastomotic region is suitable. Nevertheless, the leakage rate was not reduced. This leads to the assumption that visualizing ICG-FI alone does not predict the quality of tissue perfusion [16]. In another study, Zehetner et al performed real-time gastric tube perfusion assessments in 150 esophagectomy procedures. They found that ICG-FI was associated with a higher probability of developing a leak. The results further encouraged the presupposed correlation of sufficient blood flow and anastomotic healing [17]. These findings were supported by another clinical study [28]. They retrospectively analyzed 52 patients with mesenteric ischemia. In 11.5%, using ICG-FI led to significant modification of the surgical approach. Regardless of these promising results, in these studies, the assessment of ICG-FI was visual and therefore subjective. No objective evaluation has been carried out.

Although ICG-FI has become an increasingly common technique for evaluating intraoperative organ perfusion, only a few studies have tried to quantitatively validate the technique for visceral perfusion assessment. In an experimental setting with mesenteric ischemia in rabbits, Toens et al. measured perfusion using ICG-FI and radioactive microspheres. After defining ROIs in ischemic areas, as well as in a control area, they quantified the increase in florescence in ICG-FI. In the visual evaluation, well-perfused areas showed a strong fluorescence intensity. Regardless, the quantitative assessment showed a wide range of results. Additionally, they reported a significant rate of intra- and interindividual inconsistency.

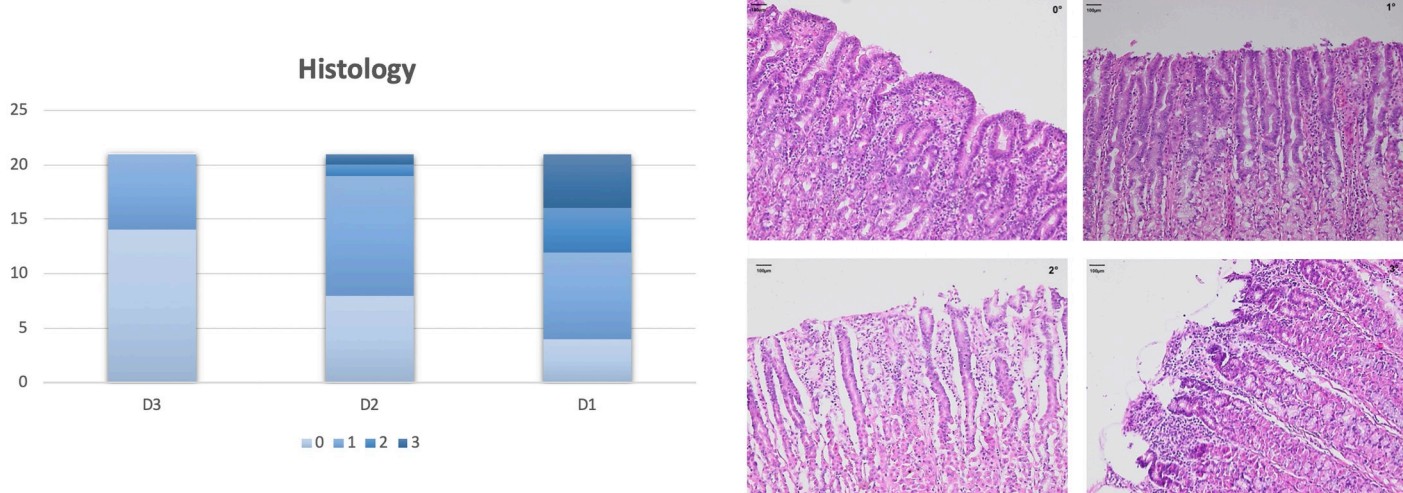

**Fig 4. Histological assessment.** Mucosa was taken from the anterior gastric corpus and divided into the following grades under a light microscope according to Zhang et al. [13]: grade 0: normal gastric mucosa; grade 1: surface mucosa cells were damaged; grade 2: in addition to extensive luminal damage, cells lining the gastric pits were also disrupted and exfoliated; grade 3: cell destruction extended into the gastric gland. On the right, an example of each grade is shown. The grade of damage increased significantly from D3 to D1 (p<0.05).

Nevertheless, they concluded that ICG-FI is practicable for the assessment of mesenterial blood flow [29].

In a porcine study of mesenteric ischemia, Diana et al. evaluated perfusion using ICG-FI. They associated their ICG-FI results with metabonomics, lactate levels, and histopathologic

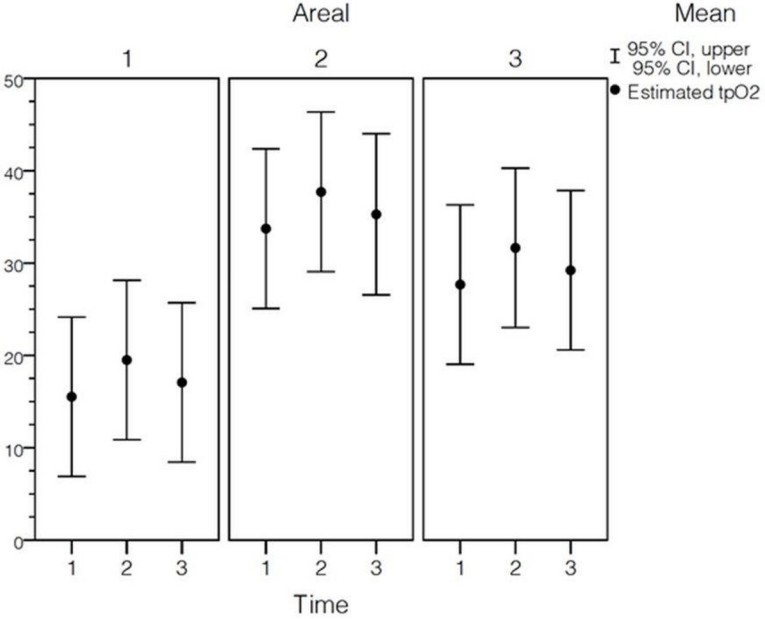

**Fig 5. Assessment of tissue oxygenation.** Flexible polarographic measuring probes (Licox, Germany) for continuous measurement of the tissue oxygen tension (partial pressure of tissue oxygen [$tpO_2$]) were inserted in the proximal (D3) and distal ends of the tube (D1). D1 shows a significant delay in the fluorescence signal at all time points ($^*$: p<0.05). P-values for fixed effects were significant for area ($p_a$), time ($p_t$), and time-dependent area ($p_{a^*t}$). D2 showed no significance (n.s.). Pairwise comparisons were based on estimated marginal means with a confidence interval of 95%.

findings. For quantification, they calculated the time to peak using the intensity curve of the different ROIs. They concluded that assessment of perfusion status using ICG-FI is feasible [17]. This study was conducted in 6 animals with an ischemic period of 1 hour. ICG-FI was only performed 15 minutes after ischemia induction. In a subsequent survival study, the group used ICG-FI to evaluate perfusion of bowel anastomoses. Thus, the grade of perfusion was calculated by their previously described technique. In 6 animals, ischemic segments were evaluated after 2, 4, and 6 hours. Two surgeons determined the resection margins according to clinical parameters. Afterwards, ICG-FI was performed and evaluated via the time-to-peak measurement. They found that in 50% of the measurements, the clinical ICG-FI evaluations were comparable after 2 and 4 hours of ischemia. This conformity was even higher after 6 hours of ischemia (75%). In discordant cases, the resection margins would have been chosen closer to the ischemic area. Consequently, using ICG-FI can prevent anastomotic failure due to impaired perfusion [30].

In this study, we studied additional methods for quantitative assessment in ICG-FI to increase the diagnostic benefit of the technique. To create a standardized experimental setting, we chose a gastric tube model. The advantage of this model is the dependence of tissue perfusion on one vessel (A. gastroepiploica dextra). Thus, falsification of results caused by collateralization is eliminated. On the other hand, this is the main disadvantage in clinical practice. Leakage of anastomosis after esophagectomy with gastric pull-up remains high due to impaired perfusion at the tip of the conduit. Based on this experimental study, we can transfer our methods to therapy for those patients.

Due to its short half-life, ICG is perfect for repeated measurements. Residual ICG storage in the tissue after several injections leads to persistent fluorescence. BSFI helps to abolish false-high values due to ICG tissue pooling by subtracting the residual florescence. As a result, multiple measurements during an operation are feasible without losing diagnostic meaningfulness. Visual interpretation of ICG-FI alone can lead to misinterpretation of perfusion status.

The SFI is defined as the maximal slope of fluorescent intensity. This parameter represents the rise of the fluorescent signal in relation to time. This value is also independent from background ICG pooling. The SFI is strongly reliant on vessel blood flow and consequently depends on cardiac output, systemic vascular resistance, volume status, and vasoconstriction.

Using the one-vessel gastric tube model, TTS could be implemented as an additional quantification technique reflecting the prolonged path from the arterial radix to the distal part of the arcade under hemodynamic change. During vasoconstriction, capillary perfusion might be reduced. Nevertheless, BSFI is not affected by hemodynamic changes such as hypo- or hypertension induced by norepinephrine, especially in the most distal areas of perfusion. While the SFI and TTS represent only the arterial flow into the capillary bed, BSFI represents the distribution of ICG in the capillary bed, reflecting the tissue perfusion itself.

ICG-FI correlated perfectly with the predicted perfusion changes. The differences in FM, as the gold standard, especially in D3 and D2, can be explained by the differences in the methods. While in ICG-FI one ROI in an area is defined, FM are evaluated as the average of a larger area. The FM technique requires a certain amount of tissue (3 g each) and several specimens, due to the complexity of the method, to obtain sufficient fluorescent spectrometric levels. Inconsistencies in the measurement of FM could result in deviations: loss of microspheres during digestion, filtering of the samples or sample handling, and spillover from each fluorescent color band to the adjacent color bands [31]. Therefore, punctual assessment for smaller ICG-FI ROIs is difficult. Tissue oxygenation reflected the lower perfusion in the distal area well. However, in contrast to ICG-FI, D2 showed improved oxygenation compared to D3. Tissue oxygenation is measured by special probes inserted into the parenchyma, resulting in

punctual measurement with probable tissue damage. Closeness to vessels results in better oxygenation, leading to better results.

The main advantage of ICG-FI is that the surgeon is able to noninvasively evaluate the perfusion in every region of interest depending on the intraoperative needs. Using the described method for quantitative assessment can further objectify ICG-FI results, leading to better prediction of tissue perfusion. This might avoid misinterpretations visual assessments only and reduce ischemic complications.

## Conclusion

ICG-FI is a feasible and promising tool for predicting visceral perfusion. The present study implies that quantitative assessment of ICG-FI data using SFI, BSFI and TTS is an effective and viable method for determining tissue perfusion. Further survival studies and clinical studies with observation of clinical outcome and determination of reference values are necessary to prove the method and make it applicable in daily practice.

## Supporting information

**S1 File. Ethic approval.**
(PDF)

## Acknowledgments

The authors gratefully acknowledge the skillful assistance of Aline Reitmeier and Jutta Dammann in organizing and supporting the animal experiments. The manuscript has been read and approved by all authors.

## Author Contributions

**Conceptualization:** Anna Duprée, Philipp H. von Kroge, Oliver Mann, Detlef Russ, Christian Detter, Sabine H. Wipper.

**Data curation:** Anna Duprée, Henrik C. Rieß, Philipp H. von Kroge, Hans O. Pinnschmidt, Sabine H. Wipper.

**Formal analysis:** Anna Duprée, Philipp H. von Kroge, Detlef Russ, Sabine H. Wipper.

**Funding acquisition:** Anna Duprée.

**Investigation:** Anna Duprée, Henrik C. Rieß, Philipp H. von Kroge, Sabine H. Wipper.

**Methodology:** Anna Duprée, Eike S. Debus, Oliver Mann, Detlef Russ, Christian Detter, Sabine H. Wipper.

**Project administration:** Anna Duprée, Jakob R. Izbicki, Eike S. Debus, Oliver Mann, Sabine H. Wipper.

**Resources:** Anna Duprée, Henrik C. Rieß, Jakob R. Izbicki, Eike S. Debus, Oliver Mann, Sabine H. Wipper.

**Software:** Hans O. Pinnschmidt, Detlef Russ.

**Supervision:** Jakob R. Izbicki, Oliver Mann, Sabine H. Wipper.

**Validation:** Anna Duprée, Hans O. Pinnschmidt, Christian Detter, Sabine H. Wipper.

**Visualization:** Detlef Russ.

Writing – **original draft:** Anna Duprée.

Writing – **review & editing:** Henrik C. Rieß, Philipp H. von Kroge, Jakob R. Izbicki, Hans O. Pinnschmidt, Christian Detter, Sabine H. Wipper.

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
