## [Decision Letter · Decision Letter 0]

27 Jul 2020

PONE-D-20-12099

Validation of quantitative assessment of Indocyanine green fluorescent imaging in an one vessel model

PLOS ONE

Dear Dr. Dupree

Thank you for submitting your manuscript to PLOS ONE. After careful consideration, we feel that it has merit but does not fully meet PLOS ONE’s publication criteria as it currently stands. Therefore, we invite you to submit a revised version of the manuscript that addresses the points raised during the review process.

The study is interesting and well conducted; however, English revision by native speaker is needed.

We look forward to receiving your revised manuscript.

Kind regards,

Diego Raimondo

Academic Editor

PLOS ONE

Additional Editor Comments:

The paper requires English correcrion by native speaker before acceptance

Journal Requirements:

2. Thank you for including your ethics statement: 'All experiments were performed in compliance with the Institutional Review Board for the care of animals in accordance with the National Institutes of Health guidelines for ethical animal research (Ref. Nr. 113/14)'.

(a) Please amend your current ethics statement to confirm that your named ethics committee specifically approved this study.

(b) Once you have amended this/these statement(s) in the Methods section of the manuscript, please add the same text to the “Ethics Statement” field of the submission form (via “Edit Submission”)

For additional information about PLOS ONE submissions requirements for ethics oversight of animal work, please refer to http://journals.plos.org/plosone/s/submission-guidelines#loc-animal-research.

3. We noticed minor instances of text overlap with the following previous publication(s), which need to be addressed:

(1) https://www.wjgnet.com/1007%C2%AD9327/full/v11/i18/2830.htm

The text that needs to be addressed involves the Results section (lines 320).

In your revision please ensure you cite all your sources (including your own works), and quote or rephrase any duplicated text outside the methods section. Further consideration is dependent on these concerns being addressed.

4. At this time, we ask that you please provide scale bars on the microscopy images presented in Figure 4 and refer to the scale bar in the corresponding Figure legend.

"This research did not receive any specific grant from funding agencies in the public, commercial, or

425 not-for-profit sectors. Cost of animals and equipment were covered by a department internal fund,

staple devices were provided by Medtronic GmbH Germany."

"The authors received no specific funding for this work. "

Additionally, because some of your funding information pertains to commercial funding, we ask you to provide an updated Competing Interests statement, declaring all sources of commercial funding.

In your Competing Interests statement, please confirm that your commercial funding does not alter your adherence to PLOS ONE Editorial policies and criteria by including the following statement: "This does not alter our adherence to PLOS ONE policies on sharing data and materials.” as detailed online in our guide for authors  http://journals.plos.org/plosone/s/competing-interests.  If this statement is not true and your adherence to PLOS policies on sharing data and materials is altered, please explain how.

Please include the updated Competing Interests Statement and Funding Statement in your cover letter. We will change the online submission form on your behalf.

6. Your ethics statement must appear in the Methods section of your manuscript. If your ethics statement is written in any section besides the Methods, please move it to the Methods section and delete it from any other section. Please also ensure that your ethics statement is included in your manuscript, as the ethics section of your online submission will not be published alongside your manuscript.

Reviewers' comments:

Reviewer's Responses to Questions

**Comments to the Author**

1. Is the manuscript technically sound, and do the data support the conclusions?

Reviewer #1: Yes

Reviewer #2: Yes

2. Has the statistical analysis been performed appropriately and rigorously? 

Reviewer #1: Yes

Reviewer #2: Yes

3. Have the authors made all data underlying the findings in their manuscript fully available?

Reviewer #1: Yes

Reviewer #2: Yes

4. Is the manuscript presented in an intelligible fashion and written in standard English?

Reviewer #1: No

Reviewer #2: Yes

5. Review Comments to the Author

Reviewer #1: In this study, the authors show that ICG-Fluorescence Imaging is a feasible, and promising tool to predict visceral perfusion in a pig model. Although the science seems sound, the paper is difficult to read and needs to be carefully edited for grammatical errors.

Reviewer #2: I read with interest this manuscript by Dupree et al. about a challenging topic in actual research: to validate a quantitative assessment of ICG perfusion of the tissues. Despite the porcine model, the method theorized could be adopted for other studies. I think this manuscript is suitable for publication in this journal.

6. PLOS authors have the option to publish the peer review history of their article (what does this mean?). If published, this will include your full peer review and any attached files.

Reviewer #1: **Yes: **Michael Bouvet

Reviewer #2: No

---

## [Author Response · Author response to Decision Letter 0]

17 Sep 2020

Dear Editor, dear Reviewers,

Thank you for suggestions concerning our manuscript. 

1)The paper requires English correcrion by native speaker before acceptance

The manuscript has been proofed and edited by the Springer Nature Author services. 

The manuscript has been adopted to the PLOS ONE Stlye requirements. 

3) Thank you for including your ethics statement: 'All experiments were performed in compliance with the Institutional Review Board for the care of animals in accordance with the National Institutes of Health guidelines for ethical animal research (Ref. Nr. 113/14)'.

(a)Please amend your current ethics statement to confirm that your named ethics committee specifically approved this study.

(b) Once you have amended this/these statement(s) in the Methods section of the manuscript, please add the same text to the “Ethics Statement” field of the submission form (via “Edit Submission”)

The ethical approvement has been amended in the Method section, and as well in the Ethic Statement section. 

4) The text that needs to be addressed involves the Results section (lines 320).

 In your revision please ensure you cite all your sources (including your own works), and quote or rephrase any duplicated text outside the methods section. Further consideration is dependent on these concerns being addressed.

We added the citation to this text. Hence, a special technique to evaluate the histology is explained, we did not rephrased it. 

5) At this time, we ask that you please provide scale bars on the microscopy images presented in Figure 4 and refer to the scale bar in the corresponding Figure legend.

We added scale bars to Figure 4. 

6) Please include the updated Competing Interests Statement and Funding Statement in your cover letter. We will change the online submission form on your behalf.

We updated the Funding Statement in the Cover letter. Thank you for your advice. 

7)Your ethics statement must appear in the Methods section of your manuscript. If your ethics statement is written in any section besides the Methods, please move it to the Methods section and delete it from any other section. Please also ensure that your ethics statement is included in your manuscript, as the ethics section of your online submission will not be published alongside your manuscript.

The Ethic statement has been added to the method section, and canceled at the end of the manuscript.

---

## [Editor Report · Decision Letter 1]

22 Sep 2020

Validation of quantitative assessment of indocyanine green fluorescent imaging in an one vessel model

PONE-D-20-12099R1

Dear Dr. Duprée,

We’re pleased to inform you that your manuscript has been judged scientifically suitable for publication and will be formally accepted for publication once it meets all outstanding technical requirements.

Kind regards,

Diego Raimondo

Academic Editor

PLOS ONE

With kind regards

Diego Raimondo

---

## [Editor Report · Acceptance letter]

29 Sep 2020

PONE-D-20-12099R1 

Validation of quantitative assessment of indocyanine green fluorescent imaging in a one-vessel model 

Dear Dr. Duprée:

I'm pleased to inform you that your manuscript has been deemed suitable for publication in PLOS ONE. Congratulations! Your manuscript is now with our production department. 

Kind regards, 

on behalf of

Dr. Diego Raimondo 

Academic Editor

PLOS ONE